# DoubleU-Net: Colorectal Cancer Diagnosis and Gland Instance Segmentation with Text-Guided Feature Control

**Abstract.** With the rapid therapeutic advancement in personalized medicine, the role of pathologists for colorectal cancer has greatly expanded from morphologists to clinical consultants. In addition to cancer diagnosis, pathologists are responsible for multiple assessments based on glandular morphology statistics, like selecting appropriate tissue sections for mutation analysis [6]. Therefore, we propose DoubleU-Net that determines the initial gland segmentation and diagnoses the histologic grades simultaneously, and then incorporates the diagnosis text data to produce more accurate final segmentation. Our DoubleU-Net shows three advantages: (1) Besides the initial segmentation, it offers histologic grade diagnosis and enhanced segmentation for full-scale assistance. (2) The textual features extracted from diagnosis data provide high-level guidance related to gland morphology, and boost the performance of challenging cases with seriously deformed glands. (3) It can be extended to segmentation tasks with text data like key clinical phrases or pathology descriptions. The model is evaluated on two public colon gland datasets and achieves state-of-the-art performance.

**Keywords:** Cancer diagnosis · Gland segmentation · Morphological feature guidance

## 1 Introduction

Colorectal cancer is among the leading causes of mortality and morbidity in the world. It is the third most common cancer worldwide (following tumors of the lung and breast), and the fourth most common cause of oncological death [22]. More than 90% of the colorectal cancers are adenocarcinomas, which are malignant tumors originating from glandular epithelium. It is determined by pathologists on Hematoxylin and Eosin (H&E) stained tissue specimens. The morphological information of intestinal glands, like architectural appearance and gland formation, is one of the primary features in clinical to inform prognosis and plan the treatment of individual patient [3]. Therefore, the automated segmentation methods that extract quantitative features with morphological statistics are essential in clinical practice to boost assessing efficiency and reliability, reducing inter- and intra-observer variability, and handling the ever-increasing image quantity and variety.

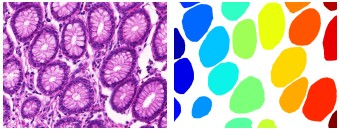 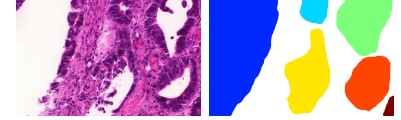

Patient 15: benign, healthy            Patient 7: malignant, moderately differentiated

**Fig. 1:** Examples of gland instance segmentation with different histologic grades (i.e. benign or malignant) and differentiation levels. Malignant cases usually show great morphological changes. Glands are denoted by different colors.

While previous approaches to address this problem focus on hand-crafted features and prior knowledges of objects [1, 5, 7, 9, 14, 19–21], recent convolutional neural networks have promoted this area by learning semantic features[4, 17, 24–26]. Besides glandular objects, the most advanced approaches focus more on capturing the boundary of the gland with different network architecture or loss function [8, 25, 26, 29, 30]. These methods show advancement in identifying clustered and touching gland objects through detailed features, but fail to cope with the morphological variance. The glandular morphology shows great diversity for different histologic grades and differentiation levels, as shown in Fig.1. The significant deformation in architectural appearance and glandular formation could undermine the robustness of methods that focus on detailed local features.

Moreover, the morphological statistics of glandular objects play an increasingly important role in clinical practice for colorectal cancer treatment, which raises higher requirements to segmentation accuracy. With the rapid development in personalized medicine, the role of pathologists has greatly expanded from traditional morphologists to clinical consultants (for gastroenterologists, colorectal surgeons, oncologists, and medical geneticists) [6]. Therefore, besides providing accurate histopathologic diagnosis as a very first step, pathologists are responsible for accurately assessing pathologic staging, analyzing surgical margins, selecting appropriate tissue section for microsatellite instability (MSI) testing and mutation analysis, searching for prognostic parameters, and assessing therapeutic effect, with the aid of quantitative features of gland [6].

To address these challenges, we propose DoubleU-Net that diagnose colorectal cancer and segment gland instance simultaneously, and utilize the diagnosis text which emphasizes the high-level features and overall structural information for more accurate gland instance segmentation. Our DoubleU-Net achieves the state-of-the-art performances in segmentation on public dataset GlaS and CRAG. Our major contributions of DoubleU-Net are listed as follows.

1. Besides the initial gland segmentation, we offer cancer diagnosis and greatly improved segmentation results as full-scale assistance for pathologists according to the clinical routine.
2. It emphasizes high-level features including the structural and morphological appearance of objects, and the significant improvement on shape similarity validates the effectiveness of textual feature guidance.

3. DoubleU-Net can well incorporate text and image from different domains for medical image segmentation. The extracted textual features are applied directly to image other than the text related tasks (e.g. label/report generation). It is also designed in a generalized way for segmentation with key clinical phrases or pathology descriptions as supplementary text input.

## 2    Related Work

**Gland Instance Segmentation.** In the last few years, various methods have been proposed for gland segmentation. Pixel-based methods [5, 14, 19, 21] and structure-based methods [1, 7, 9, 20] make full use of the hand-crafted features and prior knowledge of glandular structures. These methods achieved satisfying results on benign objects, but not for the adenoma cases with diversity in size and appearance. Recently, deep learning methods have shown remarkable performance for histological image analysis. The popular U-Net [17] is a U-shaped network with a symmetric encoder and decoder branch. Chen et al. proposed DCAN [4] that focuses on both glandular object and boundary to separate clustered glands, and won the 2015 MICCAI GlaS challenge [18]. Based on DCAN, MILD-Net [8] introduced a loss function to retain maximal information during feature extraction in training. Xu et al. [25, 26] applied a multi-channel multi-task network for foreground segmentation, edge detection, and object detection respectively. Besides, Yan et al. [29] proposed a shape-preserving loss function to regularize the glandular boundary for the gland instance. Furthermore, several methods aim at utilizing less manual annotations or computational expenses for gland instance segmentation. For example, Yang et al. [30] presented a deep active learning approach using suggestive annotation, and Zheng et al. [33] proposed a representative annotation (RA) framework. Quantization performed on FCNs [10] reduces computational requirement and overfitting while maintaining segmentation accuracy [24]. Unannotated image data can be utilized effectively by the proposed deep adversarial network [32] for considerably better results.

**Text Related Image Processing.**    There exist close relationships between language and visual attention in psychology according to recent studies [12]. This suggests that spoken or text language associated with a scene provides useful information for visual attention in the scene [13]. Mu et al. [13] introduce a text-guided model for image captioning, which learns visual attention from associated exemplar captions for a given image, referred to as guidance captions, and enables to generate proper and fine-grained captions for the image. TieNet [23] incorporates chest X-ray image with the associated medical report for disease classification and reporting. MULAN [28] is a multi-task model for lesion Detection, tagging, and segmentation. To mine training labels, Yan tokenizes the sentences in the radiological report and then match and filter the tags in the sentences using a text mining module. Similarly, the training labels are text-mined from radiology reports and further utilized in the training of lesion annotation in LesaNet [27].

It is worth noticing that the text data in these approaches are not directly applied to images, they are processed for image captioning, tag extraction, and label selection, which are related to text output or label generation. In this paper, it is novel that the features from diagnosis text data directly control the high-level visual features and the gland segmentation, which serve as the guidance on gland morphology in the learning process.

## 3    Method

Our proposed DoubleU-Net consists of two encoders for feature extraction of histological images and the corresponding text information, and two decoders [4] for pixel-wise prediction of glandular region and boundary. The architecture is similar to the shape of two Us connected at the bottom, therefore we name it DoubleU-Net. It works in the following two steps: (1) diagnose histologic grades and differentiation levels while performing initial gland segmentation, (2) the diagnosis data is fed into a text-to-feature encoder for higher level feature guidance, and the final gland instance segmentation results are obtained based on the features from different information domain.

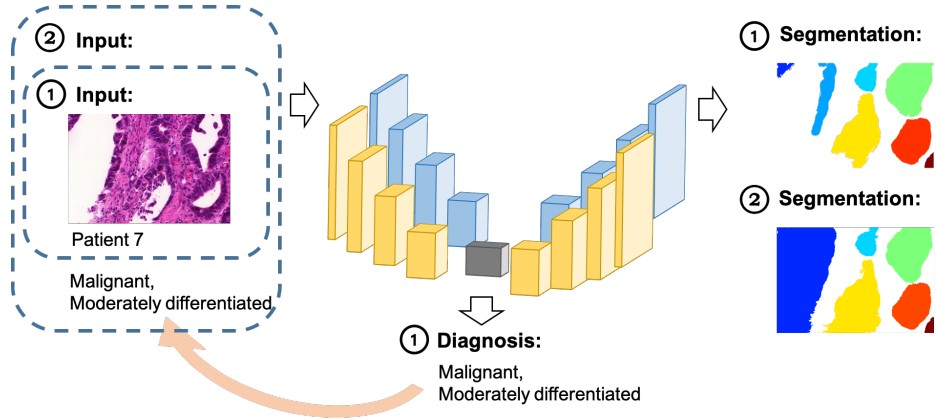

**Fig. 2:** Pipeline of the DoubleU-Net: (1) Colorectal cancer diagnosis and initial gland segmentation are predicted, and then (2) the diagnosis text data is fed into the text-to-feature encoder of DoubleU-Net, which provides morphological feature guidance and yields largely improved segmentation results.

### 3.1    Colorectal Cancer Grading

In clinical practice, the pathologists initially determine the histologic grade and differentiation level based on the glandular morphology. Similarly, DoubleU-Net

performs colorectal cancer diagnosis by utilizing the visual features, and the initial segmentation results are provided as well. In Fig.3 the grading classifier is connected to the bottom of DoubleU-Net, which consists of average pooling layers, convolutional layers, and a linear transformation layer. Besides, the feature maps from two encoders (i.e. all features maps after the max-pooling layers) are concatenated and fed into the classifier for cancer diagnosis.

The automated histologic and differentiation grading in model achieves its underlying advantages and purposes as follows: (1) Directly offer final diagnosis results with glandular morphology statistics to pathologists as reference and assistance for cancer grading. (2) The joint supervision of prediction and segmentation increases the learning ability of the model and alleviate the over-fitting during training. (3) The predicted diagnosis data (or revised data by pathologists, if possible) can further control the abstract higher-level features to improve segmentation accuracy for other treatments by pathologists, like selecting appropriate tissue sections for MSI testing and mutation analysis [6].

Besides, the gland segmentation is usually performed on image patches extracted from the original digitalized H&E stain slides (up to $10000^2$ pixels) to focus on the region of interests, and current datasets also contain image patches instead of the whole slides. Therefore, the model is required to maintain classification consistency for different images from the same patient, which is also crucial in clinical routine to perform automated segmentation and prediction on image level. Instead of appending an algorithm to unify the output grades, the patient ID of corresponding histological images can be further utilized. With patient ID as an additional input, the model is able to learn the similarity among images from the same patient, and gradually exploit the interdependency among glandular features, diagnosis grades, and patients. As shown in Table 3, text encoder with additional input significantly promotes the diagnosis accuracy in GlaS dataset. Then the predicted cancer grades and patient ID together served as text input to further refine the gland segmentation, and the grade classifier is deactivated in the remaining process.

### 3.2 Text-Guided Feature Control

The colorectal cancer is diagnosed based on the morphological statistics of colon gland instance by pathologists, and DoubleU-Net also predicts the cancer grades based on the visual features of glands. Since the diagnosis data like *moderately differentiated* suggests changes in the gland structure and appearance, we could further utilize these diagnosis data for more accurate segmentation.

**Word Embedding.** Diagnosis text data is fed into the network as an additional input to dominate high-level features from corresponding histological images, and word embedding is the very first step to represent these words as low dimensional vectors. Given limited text information for each image, the simple one-hot encoding is a straightforward option. However, it has the following limitations: (1) Most values of the one-hot encoding will be zero, which is inefficient

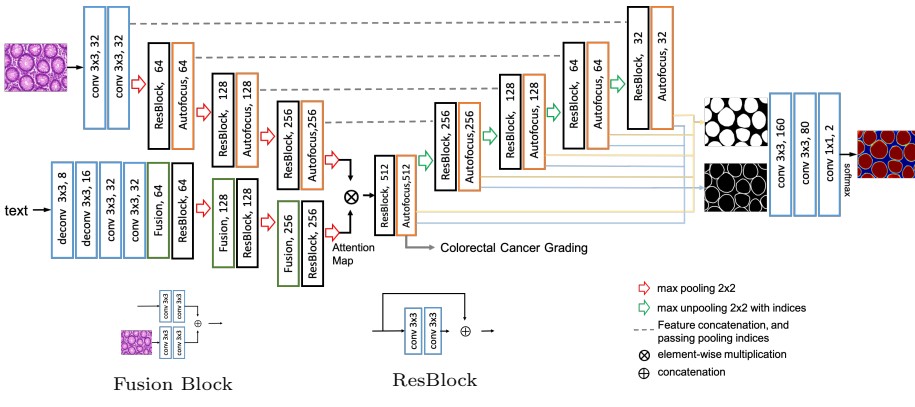

**Fig. 3:** Overall architecture of DoubleU-Net. It consists of two encoders for feature extraction of histological images and corresponding text information, and two decoders [4] for pixel-wise prediction of glandular region and boundary.

and fails to boost the learning ability of the network for text data. (2) It is not sufficient enough to reveal the semantic relationship between different phrases, because the vectors obtained by one-hot encoding are orthogonal to each other. This independent representation is suitable for histologic grade *benign* and *malignant* that have opposite meanings, but it fails to demonstrate the similarity among *benign*, *moderately differentiated* and *moderately-to-poorly differentiated*. (3) Furthermore, word embedding is a more generalized approach that can be used for a different and larger amount of text data like pathology descriptions of histological slides.

To fully capture the semantic, syntactic similarity of the word, and relation with other words, we adopt popular word2vec [11] from Google for distributed representation. This model is trained on a huge text corpus (one billion words) to construct vocabulary and learns the vector representation of words. Besides the excellent representation capability, the other advantage of word2vec is that the encoded vectors can be meaningfully combined using simple vector summation. We now formulate the word embedding of clinical data as follows:

$$v_i = \sum_j \mathcal{G}(m_{ij}; \boldsymbol{\theta}), \tag{1}$$

$$v = v_0 \parallel v_1 \parallel v_2 \parallel ... \parallel v_n, \tag{2}$$

where $\mathcal{G}$ denotes the word2vec model with parameter $\boldsymbol{\theta}$. $m_{ij}$ represents the $j$th word in $i$th sentence (or phrase in our case). $v_i$ is the corresponding vector representation of the phrase (summation of word vector representation), and all $n$ vectors are concatenated (denoted by $\parallel$) to form the textual feature $v$. In Table 1, we show the cosine distance of vectors encoded by word2vec, where histologic and differentiation grades are from several different histology images.

**Table 1:** Semantic similarity of encoded diagnosis text from GlaS dataset.

| Semantic similarity | healthy | adenomatous | moderately differentiated | moderately-to-poorly differentiated |
|---|---|---|---|---|
| benign | 0.7113 | 0.5578 | 0.5341 | 0.5273 |

**Visual and Textual Feature Fusion.** For each histological image, textual features are extracted from the corresponding encoded text data. To fully utilize the high-level clinical information for dense pixel-wise prediction, it is necessary to well incorporate the textual features with the visual ones. Therefore, the original input image is concatenated with the textual features after max-pooling and convolutional layer, as formulated in the equation below:

$$\boldsymbol{y} = \mathcal{F}(\boldsymbol{v}; \boldsymbol{\theta_\alpha}) \parallel \mathcal{F}(\boldsymbol{x}; \boldsymbol{\theta_\beta}), \tag{3}$$

where $\mathcal{F}$ denotes the convolution with corresponding weights $\boldsymbol{\theta}$. $\boldsymbol{x}$ represents the down-sample original input image, and $\boldsymbol{v}$ is the encoded text feature.

We explain the fusion block from the following three aspects. First, image and text data are from different information domains and deliver features of different levels. Without thorough incorporation, text data contains no object localization could fail to guide the visual features on the pixel level. Our multiple fusion blocks adaptively combine spatial features with appearance features from clinical text information, where local details and overall glandular morphology are gradually fused and balanced. Second, the feature integration strengthens the interdependency of glandular morphology and histologic grades, and improve the capability of the network to learn and distinguish the visual features for different cancer grades. Furthermore, combining feature maps and original input images, the network preserves detailed information that may be lost during the feature extraction.

**Feature Guidance and Attention.** The high-level features extracted from images are closely related to the glandular structure and morphology, which can be considered as a response to the histologic grades. On the other hand, the clinical diagnosis is also determined based on the architectural appearance and glandular formation. Therefore, we guide the extracted features to accurately focus on the gland morphology and structure during training, with the aid of features from known clinical text data. The emphasis on global structures and shapes enables the network to distinguish the gland instance from different cancer differentiation grades.

To provide structural information for the local feature extraction process, we employ feature maps from the text encoder as attention maps. Given a feature $\mathbf{u} \in \mathbb{R}^{C \times H \times W}$ from the text encoder, we perform two-dimensional softmax to the each channel of the feature maps, and calculate the attention weight for each spatial location ($channels, h, w$). Therefore, we are able to control the local feature with spatial attention map by element-wise multiplication. The feature

guidance by attention unit can be formulated as below:

$$y_{k,l} = \mathcal{F}(x_{k,l} a_{k,l}; \boldsymbol{\theta}), \tag{4}$$

$$a_{k,l} = \frac{exp(u_{k,l}/\tau)}{\sum_{k=1}^{H} \sum_{l=1}^{W} exp(u_{k,l}/\tau)}, \tag{5}$$

where $\boldsymbol{x} = \{x_{k,l}\}$ denotes feature from the image encoder, and $\boldsymbol{u} = \{u_{l,l}\}$ from the text branch. $\boldsymbol{a} = \{a_{k,l}\}$ represents corresponding attention probability distribution over each channel of feature map, and followed by the convolutional layer $\mathcal{F}$ with weight $\boldsymbol{\theta}$. $\tau$ is a temperature parameter. For high temperatures $(\tau \to \infty)$, all actions have nearly the same probability. For lower temperatures $(\tau \to 0^+)$, the probability of the action with the highest expected reward tends to 1, which may cause gradient issues in training.

As shown in Table 4, the increasing amount of text input effectively promotes the gland segmentation performance and especially the object-level Hausdorff distance, which validates our explanation that the clinical text data controls and emphasizes the glandular structure and morphology.

### 3.3   Loss

The task of our model varies with the two training phases of DoubleU-Net. Initially, it performs gland instance segmentation and cancer grade classification simultaneously with total loss $\mathcal{L}_{seg} + \mathcal{L}_{grading}$. Secondly, we improve the segmentation task with the incorporation of clinical text input, with loss $\mathcal{L}_{seg}$ only. The loss function of cancer grading (denoted as $g$) is an image level cross-entropy:

$$\mathcal{L}_{grading} = -\sum_{i \in \mathcal{I}} \sum_{m \in \mathcal{M}} \lambda_g \log p_{g,m}(i, y_{g,m}(i); \boldsymbol{\theta_g}), \tag{6}$$

where $\lambda_g$ is the weight of task $g$ to the total loss. For any input image $i$ in the dataset $\mathcal{I}$, $p_{g,m}(i, y_{g,m}(i))$ represents the image-based softmax classification of true labels $y_{g,c}(i)$ for class $m$ in $\mathcal{M}$. $\boldsymbol{\theta_g}$ is the weight parameters of grading task.

As for segmentation, unlike DCAN [4] that merged the final output of gland region and boundary manually, we integrate the combination step into the network by performing additional convolutional layers and softmax function. The total loss function for segmentation has three cross-entropy for different subtasks: gland foreground, gland boundary, and the final combined segmentation task (denoted as $f, b, c$ respectively). The loss function is written as:

$$\mathcal{L}_{seg} = \sum_{k \in \{f,b,c\}} \lambda_k \boldsymbol{w} \cdot \mathcal{L}_k \tag{7}$$

$$= -\sum_{k \in \{f,b,c\}} \sum_{x \in \mathcal{X}} \lambda_k w(x) \log p_k(x, y_k(x); \boldsymbol{\theta_k}) \tag{8}$$

$$w(x) = w_0 \cdot \frac{1}{max(dist(x), d_0) + \mu}, \tag{9}$$

where $\mathcal{L}_k$, $\boldsymbol{\theta_k}$ and $\lambda_k$ are the loss function, corresponding weight parameters, and coefficient of task $k$. $p_k(x, y_k(x); \boldsymbol{\theta_k})$ represents the pixel-based softmax classification at task $k$ for true labels $y_k(x)$. $x$ denotes a given input pixel in image space $\mathcal{X}$. To better identify the clustered gland instance, a weight map $\boldsymbol{w}$ is constructed in pixel-wise fashion and performed to emphasize the boundary in the segmentation task. In Equation 9, given the input pixel $x$, $dist(x)$ is the Euclidean distance from $x$ to the nearest gland. $d_0$ is the maximum distance that a pixel within $d_0$ range of the boundary will be considered. In our experiments, $w_0$ is set to 3, $d_0$ to 15 and $\mu$ equals to 1. The loss coefficient is $[1, 1, 1.5, 2]$ for task $g, f, b, c$ respectively.

Additionally, to address the variation of gland size and obtain effective receptive fields, we adopt Autofocus block [16] in our model. It consists of parallel dilated convolutional [31] branches with different rates and combined with learnable weights. In our experiments, we implement 4 branches with rates 2, 6, 10, and 14 respectively.

## 4    Experiments

We evaluated DoubleU-Net on two publicly available datasets of colon histological images: the Gland Segmentation (GlaS) dataset [18] from MICCAI challenge, and an independent colorectal adenocarcinoma gland (CRAG) dataset [8] originally used by Awan et al. [2]. Both datasets are from the University Hospitals Coventry and Warwickshire (UHCW) NHS Trust in Coventry, United Kingdom.

**Datasets and Pre-processing.** GlaS dataset is composed of 165 histological images from the H&E stained slides of 16 patients with a wide range of cancer grades (2 histologic grades, 5 differentiation levels). It consists of 85 training (37 benign(BN) and 48 malignant(MT) from 15 patients) and 80 test images (33 BN and 27 MT in Part A, 4 BN and 16 MT in Part B, from 12 patients). Most images are of size $775 \times 522$, and all the histological images are associated with instance-level annotation, corresponding patient ID, histologic grade and differentiation level. In CRAG dataset, 213 H&E images from different cancer grades are split into 173 training and 40 testing images. All images are associated with instance-level annotation and are mostly of size $1512 \times 1516$, and no text information is provided. For both datasets, we split 20% of the training images for validation during training to adjust hyperparameters. According to Graham et al. [8], both training and testing images are from different cancer grades. In our experiments, we augmented the images on the fly by performing elastic transformation, random rotation, random flip, Gaussian blur and color distortion. Eventually, we randomly cropped patches of size $480 \times 480$ as input.

**Evaluation Criteria.** We evaluated the performance of DoubleU-Net by the metrics used in MICCAI GlaS challenge from different aspects: F1 score for object detection, *object-level* Dice index for instance segmentation, and *object-level*

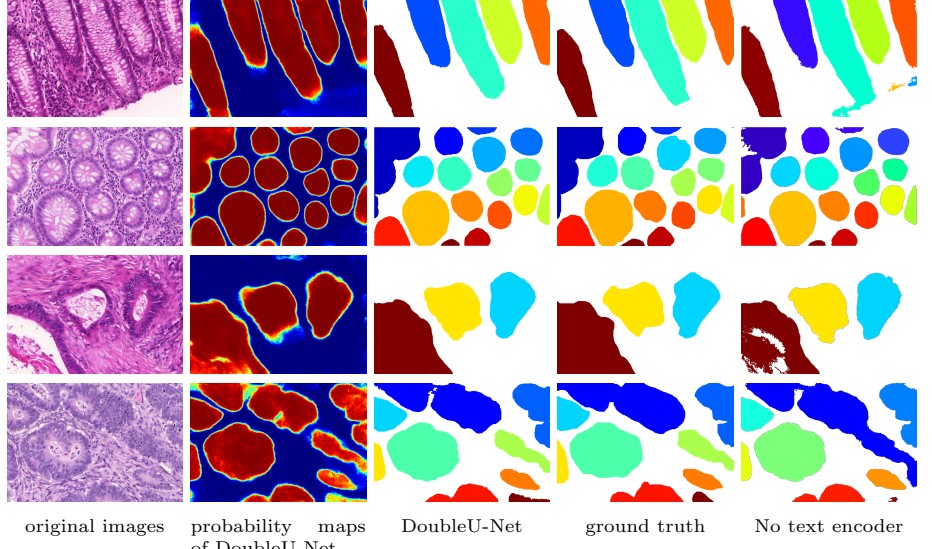

| original images | probability maps of DoubleU-Net | DoubleU-Net | ground truth | No text encoder |

**Fig. 4:** Segmentation examples of benign (top two cases) and malignant cases (bottom two cases) on GlaS dataset, and gland morphology is well recognized.

**Table 2:** Performance on GlaS dataset in comparison with other methods.

| Method | F1 Score | | Object Dice | | Object Hausdorff | |
|---|---|---|---|---|---|---|
| | Part A | Part B | Part A | Part B | Part A | Part B |
| **DoubleU-Net** | **0.935** | **0.871** | **0.929** | **0.875** | **27.835** | **76.045** |
| Mild-Net [8] | 0.914 | 0.844 | 0.913 | 0.836 | 41.540 | 105.890 |
| Quantization [24] | 0.930 | 0.862 | 0.914 | 0.859 | 41.783 | 97.390 |
| Shape Loss [29] | 0.924 | 0.844 | 0.902 | 0.840 | 49.881 | 106.075 |
| Suggestive Annotation[30] | 0.921 | 0.855 | 0.904 | 0.858 | 44.736 | 96.976 |
| Multichannel [26] | 0.893 | 0.843 | 0.908 | 0.833 | 44.129 | 116.821 |
| DCAN [4] | 0.912 | 0.716 | 0.897 | 0.781 | 45.418 | 160.347 |

Hausdorff distance for glandular shape similarity. All these evaluation metrics are conducted on gland instance instead of image level. For example, the object-level Dice index is a weighted summation of the Dice index for all the glandular objects in this image, where the weights are determined by the relative area of the glands. Equations and details can be found in the challenge paper [18].

**Implementation Details.** Our method is implemented with PyTorch [15]. We adopt Gaussian distribution ($\mu = 0, \sigma = 0.01$) for weight initialization and train with Adam optimization of initial learning rate $8 \times 10^{-4}$, with batch size of 2. We choose skip-gram architecture for the word2vec with hierarchical softmax training approach. The dimensionality of each word vector is 300 and then reshaped

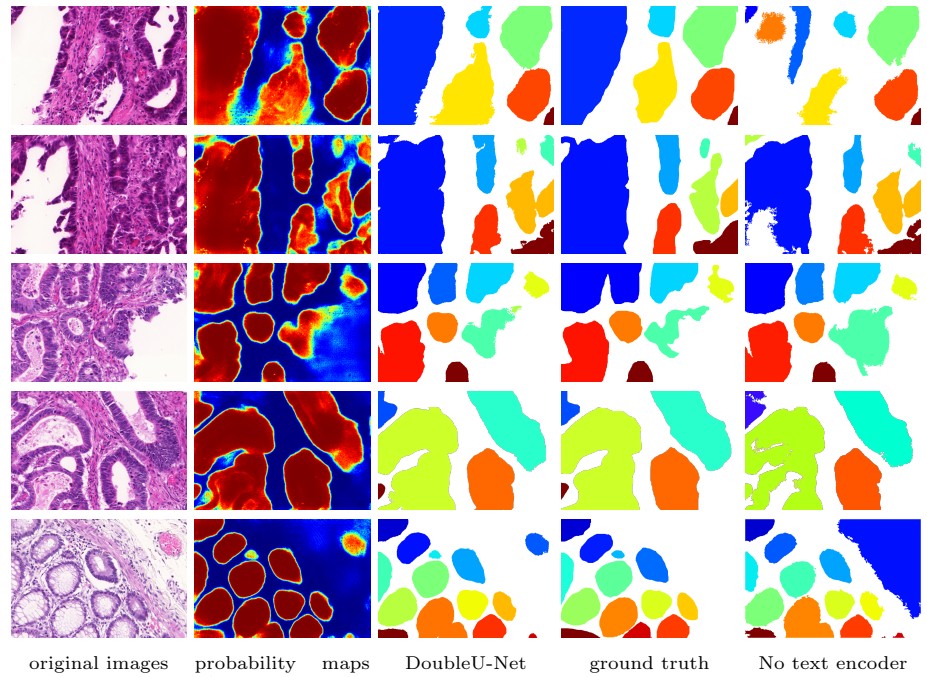

| original images | probability maps of DoubleU-Net | DoubleU-Net | ground truth | No text encoder |

**Fig. 5:** Segmentation results of DoubleU-Net on challenging cases with major improvement on GlaS dataset. DoubleU-Net with text encoder captures the structure and shape of seriously deformed glands and recognizes the misleading background. Top four rows: malignant. Bottom row: benign.

as $15 \times 20$. For cancer grading, our model infers the histologic grade based on the predicted differentiation levels, because each differentiation level is mapped to exactly one histologic grade. There is no text data in CRAG dataset (and there is no other public gland dataset with text, to the best of our knowledge), we diagnose colorectal cancer by the text encoder trained on GlaS dataset and then utilize the data for gland instance segmentation. The model is trained and tested on an NVIDIA Titan X Pascal for both datasets.

### 4.1   Results

Our DoubleU-Net achieves the best performances compared to the state-of-the-art methods on two public datasets. The morphological feature guidance from text encoder largely promotes the overall cancer diagnosis accuracy and segmentation results for seriously deformed cases.

**GlaS Dataset.** Table 2 shows the segmentation performances on GlaS dataset in comparison with the state-of-the-art methods. Among these approaches, the

**Table 3:** Accuracy of colorectal cancer diagnosis on GlaS dataset.

| Method | Histologic Grade | Differentiation Level |
|---|---|---|
| **DoubleU-Net** | **1.000** | **0.925** |
| Without Text Encoder | 0.950 | 0.725 |

**Table 4:** Ablation study of text encoder on GlaS dataset.

| Method | F1 Score | | Object Dice | | Object Hausdorff | |
|---|---|---|---|---|---|---|
| | Part A | Part B | Part A | Part B | Part A | Part B |
| Without text encoder | 0.926 | 0.857 | 0.916 | 0.859 | 41.454 | 96.473 |
| Text encoder + ID | 0.926 | 0.859 | 0.918 | 0.858 | 41.198 | 96.074 |
| Text encoder + grade 1 | 0.928 | 0.866 | 0.922 | 0.865 | 34.456 | 86.325 |
| Text encoder + grade 2 | 0.930 | 0.862 | 0.924 | 0.864 | 32.645 | 82.654 |
| Text encoder + ID & grade 1 | 0.928 | 0.868 | 0.925 | 0.866 | 33.325 | 85.435 |
| Text encoder + ID & grade 2 | **0.935** | 0.863 | 0.925 | 0.869 | 31.754 | 81.446 |
| Text encoder + grade 1 & 2 | 0.934 | 0.867 | **0.929** | 0.871 | 28.723 | 78.943 |
| **DoubleU-Net** | **0.935** | **0.871** | **0.929** | **0.875** | **27.835** | **76.045** |

quantization model [24] and Mild-Net [8] currently achieve the best results, and DCAN[4] won the MICCAI GlaS challenge 2015. Other methods focus on different aspects of gland segmentation as reviewed previously [29, 30, 26]. As shown in Table 2, our DoubleU-Net achieves the *best* results on F1, the Dice index Hausdorff distance on object-level. More importantly, DoubleU-Net outperforms other approaches by a large margin in glandular shape similarity, which validates the effectiveness of high-level feature guidance on glandular morphology and structure by abstract information from text domain. Figure 4 shows exemplary visual results from DoubleU-Net on the GlaS test images for benign and malignant cases, where glands with various structures, shapes and texture can be well-identified. The corresponding probability maps with clear glandular boundaries indicate the local details are well preserved while the high-level features are emphasized.

**Seriously Deformed Cases.** Figure 5 shows the challenging cases that are seriously deformed. The center of the deformed malignant glands is very similar to the white background in other cases. Our model without text encoder has failed to identify these white areas, and some papers presented similar results and even list these as failure cases [29, 26]. The possible reason is that the network fails to balance the local details and high-level features from a wider contextual range, and focuses more on the boundary than the overall gland structure. Our DoubleU-Net emphasizes exactly on the glandular morphology and appearance, and successfully identify these misleading areas and deformed glands with high confidence as shown in the probability maps. Besides, based on our analysis of the experimental results, the success on extremely complex cases (like Figure 5) contribute a major improvement to the overall results of DoubleU-Net, especially the object-level Hausdorff distance.

**Table 5:** Performance on CRAG dataset in comparison with other methods.

| Method | F1 Score | Object Dice | Object Hausdorff |
|---|---|---|---|
| **DoubleU-Net** | **0.835** | **0.890** | **117.25** |
| Mild-Net[8] | 0.825 | 0.875 | 160.14 |
| DCAN[4] | 0.736 | 0.794 | 218.76 |

**Colorectal Cancer Grading.** Table 3 presents classification results on histologic grade and differentiation level on GlaS dataset. With the text encoder and additional patient ID, the model achieves 100% accuracy on the histologic grade and 92.5% on the differentiation level. Therefore, DoubleU-Net is aware of the interdependencies among images from the same histological slide and maintain classification consistency for images from the same patient. Besides, due to the number of classes, it is more challenging for both methods to identify the differentiation level than the binary histologic grade.

**Ablation Study.** Table 4 shows the effectiveness of text encoder on improving the gland segmentation on GlaS dataset. Based on the network architecture of Double-Net, we gradually feed more text data and evaluate the performances. In Table 4, ID denotes the 16 patient IDs (i.e. 1 to 16) of histological slides; grade 1 means the 2 histologic grades (i.e. *benign* and *malignant*); grade 2 represents the 5 differentiation levels (i.e. *healthy, adenomatous, moderately differentiated, moderately-to-poorly differentiated, and poorly differentiated*). Comparing to DoubleU-Net without text encoder, the attention maps from the textual features significantly boosts the performance in all three evaluation metrics. We find out the patient ID brings more improvement on the classification task (Table 3) over segmentation, which by maintaining consistency and building interdependency for the images of the same slide. Both two diagnosis grades associated with gland appearance promote the segmentation results evidently, and especially on Hausdorff distance that measures the shape similarity. The slight advantage of grade 2 over grade 1 is probably because of the more detailed classification criteria brings more information into appearance features. With text encoder and all text data involved, we achieve the best gland segmentation performance with remarkable improvement.

**GRAG Dataset.** Table 5 shows the segmentation performance on CRAG dataset. The dataset is recently released by Graham et al. [8], and we report their segmentation results together with DCAN [4]. Similarly, DoubleU-Net achieves the best results on the F1 score and object-level Dice index, and a major advancement in Hausdorff distance because of the control on the overall glandular structure. As mentioned earlier, there is no text description or diagnosis information in CRAG dataset, and we use the text encoder trained on GlaS dataset to determine the histologic grades and then utilize them as feature guidance. Despite all this, DoubleU-Net still manages to promote the glandular shape similarity.

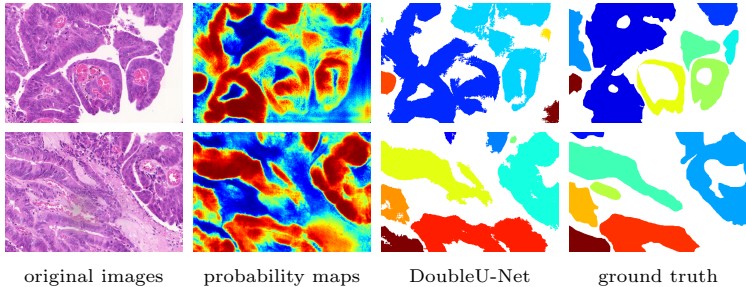

original images    probability maps    DoubleU-Net    ground truth

**Fig. 6:** Failure segmentation results by DoubleU-Net on GlaS dataset.

In Figure 6 we also investigate some poorly segmented cases by DoubleU-Net. Besides the extreme complexity of these gland structures, there are limited similar training samples in the dataset, which could be the major reason for the failure. In addition, our DoubleU-Net outputs some unclear or rough glandular boundaries, and some connecting glands for some challenging cases, appropriate post-processing or regularization on the edges can be further considered.

**Discussions.** The pathologists diagnose cancer based on the morphology of the gland. Since the diagnosis data like *moderately differentiated* suggest changes in the gland structure and appearance, we could further utilize the diagnosis data for more accurate segmentation. The diagnosis accuracy and segmentation performance validate the effectiveness of textual features. Without given clinical text data, our model also demonstrates its robustness by enhancing the segmentation performance based on predicted cancer grades in CRAG dataset. This work is an attempt to incorporate information from different domains for segmentation task (current methods utilize text for label/report generation that is not directly applied to images), and achieves the best performances because of the significant improvement in very complicated and misleading cases. We hope to see more approaches that make sufficient use of clinical text data or even other types of information for accurate medical image segmentation in future.

## 5    Conclusion

We establish a pipeline to offer the pathologists initial segmentation statistics, histologic grades, differentiation levels, and greatly improved segmentation results for full-scale assistance in the clinical assessment routine of colorectal cancer. Applying features extracted from diagnosis text data to visual cues directly, our proposed DoubleU-Net effectively guides and controls the extracted high-level features to the precise gland structure and morphology. With a major improvement on the extremely deformed and misleading cases, we achieve the best performances among the state-of-the-art methods on the two publicly available colon gland datasets and a significant advancement in object-level Hausdorff distance.

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
