# OpenReview forum: "DoubleU-Net: Colorectal Cancer Diagnosis and Gland Instance Segmentation with Text-Guided Feature Control"
_thecvf.com/ECCV/2020/Workshop/BIC — BIC 2020 Oral_

### Official Review · AnonReviewer2 · 2020-07-30
**Difficult to read paper and thus difficult to understand, but with an interesting approach and impressive results.**

**Rating:** 6
**Confidence:** 4

**Review:**

### Summary
- The authors propose a new approach called DoubleU-Net for gland instance segmentation in the context of cancer diagnosis. The classification results (different stages of cancer) from the first phase are reused to enhance the instance segmentation in a second phase. The doubleU-Net has a text-image encoding path, that incorporates the classification results in form of text (word2vec) and combines it with the original image to finally produce an attention map that is applied to the bottom of the U-Net for instance segmentation.

- The authors can show that their approach indeed improves the instance segmentation results greatly when compared to state-of-the art methods and their own method without using the text-encoding path.

### Major strengths of the paper
- The results are impressive and are better than current state of the art. The authors can clearly demonstrate that segmentation results improve when intermediate classification results (text) are presented to the network in a second iteration.

- The text-(image)-encoding path of the author's proposed method is flexible such that it can potentially be enriched with more meta-data about the patient. Their research thus goes in an important direction, that is very relevant and not explored yet in the field.

### Major Weaknesses of the paper
- The introduction and motivation is difficult to understand, and written in a very difficult and cumbersome language (see language section for more constructive feedback).

- It would be important to stress out that the presented model even without using a text input already outperforms most state-of-the-art methods. To make this more visible in the paper, results from Table 4 row “Without Text encoder” could also be added to the summary results Table 2.

- The method is not described in full detail or it is difficult to understand. There are thus still some unclear parts about how the method actually works:
  1. In the method section, it is not described how to get from the output of the network (a softmax map) to the final instance segmentation (each object being represented with a different ID).
  2. There are two training iterations (1) and (2) (described in Figure 2), but it is unclear whether in training iteration (2), the image-encoding path is trained from scratch or retrained (eg. weights are initialized from training (1)).
  3. For cancer classification/grading (results Table 3), is the network after training phase (1) or (2) used? If (1), why do results improve with text encoding? If (2), how can grading get better if during phase (2), the network is only trained on L_seg and not the grading loss anymore?
  4. Word embedding: Based on the dataset, I assume that there is a very limited amount of words for each image (and I can’t recognize any sentences), but in the section about word embedding, an embedding is described that involves the summation of multiple words for each sentence and the combination of multiple sentences. It is also unclear how the final textual feature vector v (whose length depends on n) always meets the length of 300 described in section 4. Implementation details (--> The dimensionality of each word vector is 300).

### Detailed suggestions
Is equation (3) (fusion of images and text) done once or multiple times? It would be easier for the reader, if the paper says that equation 3 is repeatedly used in the encoding path of the text. Additionally, to improve clarity, you could add a figure reference into the text (for instance to line 280): this fusion block is displayed in green in Figure 3. It is probably misleading if the text-encoder is called text-encoder, since both text and image are encoded.

### Language
- "It is determined by pathologists on Hematoxylin and Eosin (H&E) stained tissue specimens."
  Preposition "on" seems wrong / or unclear sentence.

- "The morphological information of intestinal glands, like architectural appearance and gland formation, is one of the primary features in clinical to inform prognosis and plan the treatment of individual patient."
  in clinical seems wrong

- "Besides the initial gland segmentation, we offer cancer diagnosis and greatly improved segmentation results as full-scale assistance for pathologists according to the clinical routine."
  as full-scale assistance seems wrong

- "Unannotated image data can be utilized effectively by the proposed deep adversarial network [32] for considerably better results."
→ Unlabeled image data

- Line 129
  model for lesion Detection, tagging, and segmentation.
→ model for lesion detection, ... (no capital letter)

- Line 289 "Without thorough incorporation, text data contains no object localization could fail to guide the visual features on the pixel level."
  Unclear sentence;

- Line 311 "from the text encoder, we perform two-dimensional softmax to the each channel of the feature maps"
  Remove “the”

- Line 330 "which validates our explanation that the clinical text data controls and emphasizes the glandular structure and morphology."
  Unclear sentence


**Reviews Visibility:**

I agree that my anonymized review is made publicly visible, if the submission is accepted.

---

### Official Review · AnonReviewer1 · 2020-07-30
**Solid work, great approach, good results**

**Rating:** 9
**Confidence:** 4

**Review:**

I think the presented approach of improving segmentation results of glands in histopathological images by additionally taking clues from textual descriptions is a great idea and the proposed method is indeed improving compared to a baseline not using the additional text.
The proposed method does, in fact, reach state-of-the art performance on two well known public datasets.

The paper is at times a bit confusing (e.g. numberings in Figure 2 and caption is confusing me a lot, or the floats are at quite far away from the main text talking about them) and expects the reader to know quite a lot about the medical problem at hand (lots of expert lingo). Also the technical parts are not leaving me with total clarity. If this was not a workshop paper I would value these issues significantly higher.

Anyway, I liked the paper, it is absolutely hitting the BIC spirit, and I would love to hear a talk about this work.

**Reviews Visibility:**

I agree that my anonymized review is made publicly visible, if the submission is accepted.

---

### Decision · Program_Chairs · 2020-07-31

Accept (Oral)